# The impact of serum potassium ion variability on 28-day mortality in ICU patients

YuChou Zhang[1], ShengDe Liang[2], HanChun Wen[1]*

1 Intensive Care Medicine Department, The First Affiliated Hospital of Guangxi Medical University, Nanning, China, 2 Plastic Surgery, The First Affiliated Hospital of Guangxi Medical University, Nanning, China

* nnawen@163.com

## Abstract

### Objective

Potassium ion disorders are prevalent among patients in Intensive Care Units (ICUs), yet there is a notable deficiency in established protocols and supplemental plans for potassium management. This retrospective study conducted at a single center aims to explore the relationship between potassium levels, their variability, and the 28-day mortality rate in ICU patients.

### Methods

This study analyzed data from patients admitted to the ICU of the First Affiliated Hospital of Guangxi Medical University between October 2022 and October 2023. We assessed serum potassium variability using the coefficient of variation and categorized it into four quartile groups (Q1, Q2, Q3, Q4). Additionally, patients were classified into six groups based on serum potassium concentrations. The associations between these categories and the 28-day mortality rate were evaluated using binary logistic regression, adjusting for potential confounders.

### Results

A total of 506 patients and 12,099 potassium measurements were analyzed. The group with the lowest potassium variability (Q1) exhibited the lowest mortality rate at 21% (P<0.01). It is noteworthy that within 28 days in the intensive care unit (ICU), the coefficient of variation (CV) of potassium levels significantly increased among deceased patients compared to surviving patients (P < 0.01).

### Conclusion

Significant variability in potassium levels is associated with an increased risk of 28-day mortality among ICU patients, underscoring the need for stringent monitoring and management of potassium levels in this population.

**Data Availability Statement:** All files are available from the Dryad. https://datadryad.org/stash/share/KqpjmL0KNcXqgOnPNpw-1BqHXBrBxHoxrTdU3TUzlUI.

**Funding:** The author(s) received no specific funding for this work.

**Competing interests:** The authors have declared that no competing interests exist.

## Background

ICU patients commonly manifest electrolyte imbalances, with potassium ion disturbances being prevalent [1–3]. Potassium, one of the body's three major ions, assumes a crucial role in cellular metabolism, osmotic pressure regulation, acid-base balance preservation, and modulation of neural and muscular processes [4]. Onset of hypokalemia or hyperkalemia often results in cardiac rhythm disturbances, potentially leading to sudden cardiac arrest in severe cases [5]. Consequently, both hyperkalemia and hypokalemia can significantly impact patients' physiological state and prognosis. Although approximately 98% of potassium resides within intracellular fluid in the human body, only about 2% is present in the extracellular fluid [6]. Nevertheless, with regard to extracellular potassium, the human body exhibits a sophisticated mechanism to uphold equilibrium in potassium ion concentration. This mechanism encompasses renal regulation, transmembrane transport of potassium ions, and potassium excretion functions in the intestines and skin [6]. Rigorous control is exercised over blood potassium ion concentrations to maintain them within the accepted normal range, currently set at 3.5 to 5.5 mmol/L [7, 8]. In cases of ICU hyperkalemia or hypokalemia, swift adjustment of potassium concentration is essential to preserve normal cell membrane electrophysiology [9]. Research indicates that maintaining potassium levels within the range of 3.5–4.5mmol/L can significantly reduce mortality rates among critically ill patients [10]. The 2004 American College of Cardiology/American Heart Association (ACC/AHA) guidelines further advocate for a potassium concentration exceeding 4.0 mmol/L to diminish the risk of ventricular fibrillation. Current studies predominantly address the impact of potassium concentration on mortality, with limited investigation into the effects of potassium fluctuations [11]. Recent evidence, however, suggests a correlation between the trajectory of potassium level changes and all-cause mortality, highlighting a potential association between potassium variability and mortality in critically ill patients [12]. Despite this, research on the link between potassium variability and mortality remains scant, leading to uncertainties about how serum potassium variability influences prognosis. This study seeks to elucidate the relationship between potassium variability and 28-day mortality rates.

## Methods

### Research design

This study was conducted as a descriptive observational analysis at a single center. We compared the general characteristics of the participants and examined the data characteristics between the groups of patients who survived and those who did not. Potassium levels were categorized into two groups: the Potassium Variability Group and the Potassium Concentration Group. In the potassium variation group, patients were divided into four quartiles (Q1, Q2, Q3, Q4) based on the potassium variation coefficient within 28 days of ICU. The potassium concentration group was categorized into six groups based on the average potassium concentration within 28 days of ICU admission: <3.5, 3.5–4.0, 4.0–4.5, 4.5–5.0, 5.0–5.5, and >5.5 mmol/L. The association between potassium variability and 28-day mortality was analyzed using binary logistic regression, adjusting for potential confounders. Additionally, the relationship between potassium variability magnitude, mortality across different potassium concentrations, and the changes in potassium ion concentration and coefficient of variation within the 28 days were evaluated. Patients were also segregated into the Abnormal Potassium Concentration Group (potassium concentration <3mmol/L or >6mmol/L during ICU stay) and the Normal Potassium Concentration Group (potassium concentration between 3-6mmol/L during ICU stay), to analyze the variability change within these groups over the 28 days.

## Inclusion and exclusion criteria

Inclusion criteria were specified as individuals aged 18 years or older who had been admitted to the ICU for a period exceeding two days. Exclusion criteria include patients with potassium concentration data loss of more than 50% within 28 days of ICU admission, patients with general data loss of more than half, patients with ICU stay of less than two days, and patients whose clinical outcomes (survival or death) are still uncertain at 28 days.

## Data collection

Within 28 days of admission to the ICU, potassium levels were systematically measured every 12 hours. Data collection commenced from the first day of the patient's ICU stay, encompassing a comprehensive range of variables. These variables included demographic information (age, gender), the duration of the hospital stay, clinical outcomes at 28 days, heart rate, systolic and diastolic blood pressure, mean arterial pressure, the Sequential Organ Failure Assessment (SOFA) score, pH levels, eGFR, creatinine, bilirubin, blood glucose, urine output, and the administration of insulin, potassium chloride, and furosemide. For patients who die within 28 days, if the blood drawn before death happens to occur within 12 hours, the abnormal blood drawn results will be discarded and then incorporated into other blood drawn results at similar times.

This approach was adopted to ensure the integrity of the data and to minimize the influence of external factors.

## Research period and population

The study was conducted among patients admitted to the Intensive Care Unit (ICU) of the First Affiliated Hospital of Guangxi Medical University over a one-year period, from October 2022 to October 2023. Ultimately, the study encompassed a cohort of 506 patients.

## Definition of potassium variability

The variability of serum potassium ion concentration is defined by the coefficient of variation, which is calculated as the ratio of standard deviation to the mean serum potassium ion concentration and expressed as a percentage (coefficient of variation = standard deviation/mean x 100%) [13]. A higher coefficient of variation signifies increased fluctuations in serum potassium ion levels, denoting greater variability.

## Definition of disease

ICD-10-diagnoses were determined by physicians depending on the guidelines valid at the time. ICD-10-GM derived covariates are as followed: Coronary Heart Disease I25.103; Cardiac Dysfunction I50.900- I50.907, I50.101; Hypertension I10.x00; Cerebrovascular Diseases I61.400, I61.802, I61.004, I61.300; Chronic Obstructive Pulmonary Disease J44.000, J44.100, J44.900; Diabetes E10.900, E11.900, E14.900; Renal Insufficiency N18.001-N18.004; Shock R57.200, A41.900. Potassium levels greater than 6mmol/L are considered a critical state of hyperkalemia. This may be life-threatening and therefore requires urgent reduction of potassium [14]. This measure may have an impact on potassium variability, therefore we define hyperkalemia as potassium>6mmol/L and hypokalemia as potassium<3mmol/L.

## Statistical methods

For normally distributed quantitative data, T-tests and Analysis of Variance (ANOVA) were utilized to evaluate the significance between patient outcomes (survived vs. deceased), whereas

non-parametric tests were applied for non-normally distributed data. Descriptive statistics were employed to delineate baseline clinical and demographic characteristics. Additionally, a comparative analysis between the survival and mortality groups was conducted. To elucidate the variability in potassium ion concentrations over a 28 day period, the daily coefficient of variation was calculated and plotted. To examine the association between potassium ion variability and 28-day mortality, binary logistic regression analysis was employed. Initially, the analysis explores associations between various variables and 28-day mortality, subsequently selecting appropriate variables for model inclusion. Differences among potassium concentration groups (<3.5, 3.5–4.0, 4.0–4.5, 4.5–5, 5–5.5, >5.5 mmol/L) within potassium variability quartiles (Q1, Q2, Q3, Q4) were analyzed and depicted in a histogram.

Comprehensive sensitivity analyses were conducted to validate the robustness of our data processing and resultant outcomes. To minimize the influence of potential outliers, patients with serum potassium ion concentrations below 3.0 mmol/L or above 6.0 mmol/L during hospitalization were empirically excluded [14]. Subsequently, the associations were reassessed using binary logistic regression analysis.

Considering that blood purification may have an impact on potassium variability, we excluded patients with blood purification and reanalyzed the relationship between potassium variability and mortality.

eGFR, The potential effects of urine output, blood purification, Glucose, and pH on serum potassium ion concentration were also included in the model. Furthermore, due to the potential impact of insulin, potassium chloride, and furosemide on serum potassium ion levels, the initial dose was included as a variable in the model.

To mitigate the potential impact of missing patient data, a variety of interpolation techniques were applied to potassium values at twelve-hour intervals throughout the 28 days ICU admission period. Multiple interpolations were conducted using SPSS version 27.0. The Markov Chain Monte Carlo (MCMC) method was utilized to interpolate missing values, relying on the distribution of other variables in the dataset. A continuous iterative interpolation method, with a total of ten iterations, was employed for the interpolation process. Upon completion of the interpolation, five sets of interpolated data were obtained, from which one set was randomly selected as the final interpolation result. The interpolated data were then re-analyzed.

Statistical analysis and data processing were performed using IBM© SPSS© Statistics Edition 27, Microsoft® Excel 2021 developed by Microsoft Corporation, and GraphPad Prism version 10.1.0 by GraphPad Software.

## Ethics

This study strictly adhered to the principles outlined in the Helsinki Declaration, ensuring alignment with established medical ethics standards. Approval for the research was obtained from the Medical Ethics Committee of the First Affiliated Hospital of Guangxi Medical University (Approval Number: 2024-E059-01). Due to the retrospective study design, the IRB of the Medical Ethics Committee of the First Affiliated Hospital of Guangxi Medical University waived the requirement of informed consent of study subjects. Importantly, the research did not entail the collection or storage of human tissue.

## Results

### General characteristics of study patients

This study encompasses data from 506 patients, incorporating a comprehensive collection of 12,099 serum potassium measurements results obtained within 28 days of ICU. Subjects with

incomplete data were excluded, as detailed in S1 Appendix. Among these measurements, a total of 6,166 values were missing. To address these missing values, a multiple imputation method was applied, ensuring the robustness and consistency of subsequent comparative analyses, as elaborated in S2 Appendix. Demographic and clinical characteristics of the study population are detailed in Table 1. The cohort, predominantly middle-aged, had an average age of 58 years, and the average Sequential Organ Failure Assessment (SOFA) score recorded was 8 points.

**Table 1. General information and characteristics of study participants general information and characteristics of study participants.**

|  | Average (range) |
|---|---|
| Total CV Of Potassium(%) | 12(2~45) |
| Total SD Of Potassium | 0.51(0.05~2.64) |
| Heart Rate, beats per minute | 100(46~194) |
| Systolic Pressure, mmHg | 129(50~221) |
| Diastolic Pressure, mmHg | 73(33~139) |
| Mean Arterial Pressure, mmHg | 91(41~166) |
| Potassium Minimum, mmol/L | 3.24(1.7~4.6) |
| Potassium Maximum, mmol/L | 5.22(3.2~9.6) |
| Hyperkalemia, n(%) | 74(15) |
| Hypokalemia, n(%) | 119(24) |
| Potassium Average, mmol/L | 4.14(3.15~6.8) |
| Male, n(%) | 363(72) |
| Age, years | 58(18~101) |
| Hospitalization Days | 18(2~99) |
| SOFA | 8(2~18) |
| Glucose SD | 1.9(0~13.23) |
| Glucose CV(%) | 20(0~73) |
| Glucose Average, mmol/L | 9.36(3~24.19) |
| Glucose Maximum, mmol/L | 11.76(0~41.6) |
| Glucose Minimum, mmol/L | 6.53(0~17.1) |
| Urine output, ml | 1493(0~5700) |
| Hemodialysis, n(%) | 121(24) |
| eGFR | 96(2.5~589) |
| Insulin, u | 14.35(0~124) |
| Furosemide, mg | 9.1(0~110) |
| Potassium Chloride, g | 1.2(0~12) |
| PH | 7.41(6.85~7.63) |
| Oxygenation Index, mmHg | 228(31~649) |
| Creatinine, μmol/L | 155(16~1705) |
| Bilirubin, μmol/L | 31(2.1~378) |
| Platelet, 10*9/L | 205(2~3206) |
| Coronary Heart Disease, n(%) | 54(11) |
| Cardiac Dysfunction, n(%) | 35(7) |
| Hypertension, n(%) | 126(25) |
| Cerebrovascular Diseases, n(%) | 30(6) |
| Chronic Obstructive Pulmonary Disease, n(%) | 8(2) |
| Diabetes, n(%) | 65(13) |
| Renal Insufficiency, n(%) | 218(43) |
| Shock, n(%) | 177(35) |

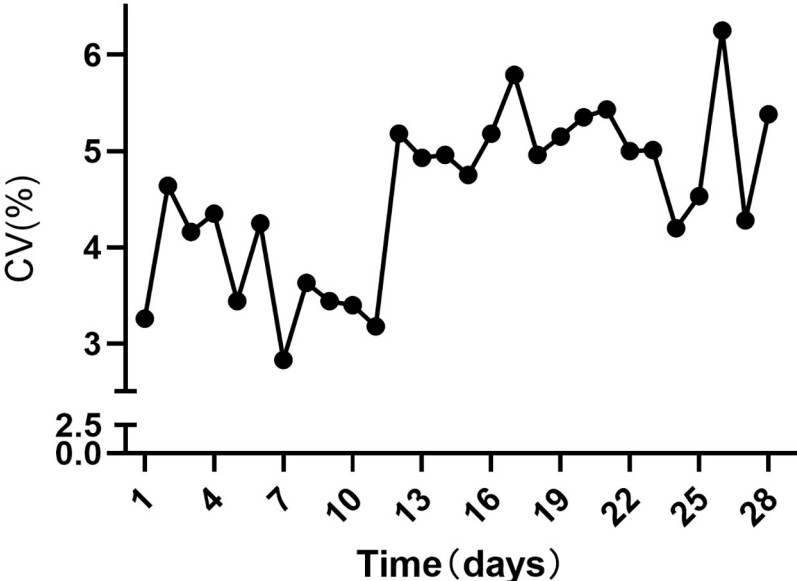

**Fig 1.**

## A general overview of serum potassium

During their ICU stay, 74 patients exhibited elevated potassium levels ($> 6$ mmol/L), while 119 demonstrated low potassium levels ($< 3$ mmol/L). The average serum potassium concentration within 28 days of ICU was 4.14 mmol/L. Fig 1 illustrates a gradual increase in average potassium variability within 28 days of ICU. S5 Appendix details a significant difference in the incidence of hyperkalemia ($> 6$ mmol/L) between the death and survival groups ($P<0.01$). Additionally, the death group was more susceptible to hyperkalemia, had elevated average potassium levels, longer ICU days, and an increased likelihood of shock ($P<0.05$).

## Potassium concentration

During the 28 day stay at ICU, the average potassium concentration predominantly fluctuated between 3.5 and 5 mmol/L, with the lowest mortality rate (31.5%) observed in patients with potassium levels between 3.5 and 4 mmol/L. Fig 2 demonstrates that mortality rates increase with greater variability in potassium concentrations among different groups. Additionally, within the potassium concentration range of 3.5 to 5 mmol/L, an increase in potassium levels corresponded to increased mortality, as documented in Table 2 of S3 Appendix. Among groups with varying potassium levels ($<3.5$, 3.5–4.0, 4.0–4.5, 4.5–5, 5–5.5, $>5.5$ mmol/L), the group with the lowest potassium variability (CV1) had a lower mortality rate compared to other potassium variability groups (CV2, CV3, CV4), as shown in Table 3 of S3 Appendix. In addition, within the potassium concentration range of 3.5 to 5 mmol/L, an increase in potassium levels corresponds to an increase in mortality rate, as shown in Table 2 of S3 Appendix.

## Variability of serum potassium

The average coefficient of variation (CV) of serum potassium levels within 28 days of ICU is 12%, ranging from 2% to 45%. In the binary logistic regression analysis, variables from the unadjusted univariate analysis with a P value $< 0.1$, which included both survival and death

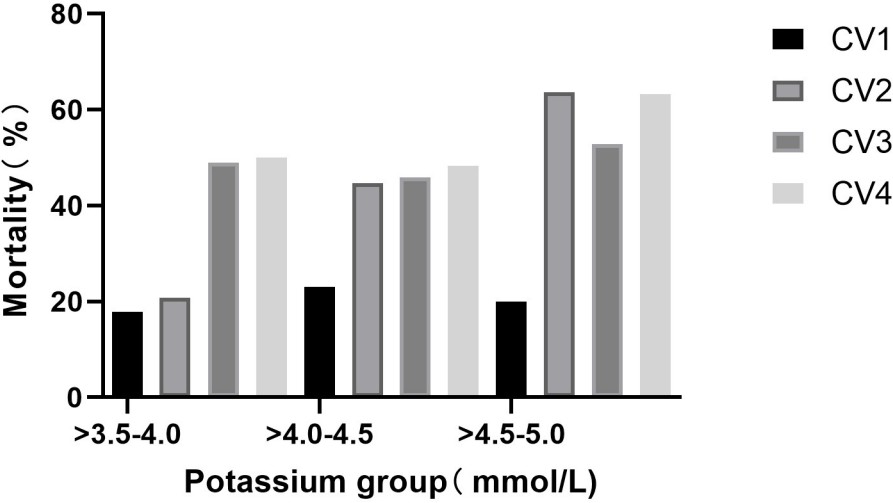

**Fig 2.**

groups, were considered. Group Q1 was established as the baseline for comparisons with the other three groups (Q2, Q3, and Q4). In the binary logistic regression model, groups Q2, Q3, and Q4 showed significant differences when compared to group Q1 (P < 0.01). After adjusting for confounding factors, including sex, age, and ICU duration, the analysis revealed a significant association between increased quartile serum potassium variability and higher mortality risk (P < 0.01). Additionally, individuals in the highest quartile of serum potassium variability exhibited a significantly increased mortality risk, with an odds ratio of 5.4 (95% CI 3.0–9.7), compared to those in the lowest quartile, as detailed in Table 2.

We then analyzed the changes in the coefficient of variation (CV) over 28 days between the death and survival groups, finding that the CV was higher in the death group than in the survival group (Fig 3). Within 28 days in the ICU, the potassium coefficient of variation (CV) was significantly higher in patients who did not survive compared to those who did survive, as evidenced by a p-value of less than 0.01 (S5 Appendix).

We divided the samples into two groups based on potassium concentrations: the abnormal group (potassium concentration>6 mmol/L or <3 mmol/L) and the normal group (potassium concentration between 3 and 6 mmol/L). The trend of CV changes over 28 days was plotted, revealing that the CV was higher in the abnormal potassium concentration group compared to the normal group (Fig 3 in S4 Appendix).

In the analysis of different potassium concentration groups (<3.5, 3.5–4.0, 4.0–4.5, 4.5–5, 5–5.5, >5.5 mmol/L), 94% of the samples were concentrated within the 3.5–5 mmol/L range. The lowest mortality rate (31.5%) was observed in the 3.5–4 mmol/L group (Table 2 in S3 Appendix). Subsequent analysis of the coefficient of variation (CV) across different potassium concentration groups revealed that mortality rates increased with greater potassium variability within the 3.5–5 mmol/L range (Fig 2 in S3 Appendix). To further explore the relationship between potassium concentration and mortality, we segmented the potassium concentration into three groups (3.5–4.0, 4–4.5, 4.5–5 mmol/L), using the 3.5–4 mmol/L range as the baseline for comparison with the other two groups. In the binary logistic regression model, 28-day mortality served as the dependent variable, with potassium concentration groups acting as independent variables. The adjusted analysis showed that, compared to the 3.5–4.0 mmol/L group, the 4.5–5 mmol/L group exhibited significantly higher mortality (P < 0.01). No

**Table 2. Related statistical models.**

| | Death, n(%) | Model1 | | Model2 | |
|---|---|---|---|---|---|
| | | OR (95%CI) | P-VALUE FOR TRENT | OR (95%CI) | P-VALUE FOR TRENT |
| Q1(CV≤9.17%) | 27(21) | 1.00(Reference) | — | 1.00(Reference) | — |
| Q2(9.17%<CV≤11.43%) | 51(40) | 2.85(1.59~5.09) | P<0.01 | 3.01(1.66~5.45) | P<0.01 |
| Q3(11.43%<CV≤14.37%) | 63(49) | 4.39(2.49~7.75) | P<0.01 | 4.87(2.7~8.77) | P<0.01 |
| Q4(CV>14.37%) | 70(54) | 5.12(2.9~9.05) | P<0.01 | 5.4(3.02~9.66) | P<0.01 |
| Covariates | | | | | |
| Male | | | | 1.53(1~2.34) | P<0.05 |
| Age | | | | 1.02(1.01~1.03) | P<0.01 |
| ICU hospitalization days | | | | 0.98(0.96~0.99) | P<0.01 |
| | Death, n(%) | Model3 | | Model4 | |
| | | OR (95%CI) | P-VALUE FOR TRENT | OR (95%CI) | P-VALUE FOR TRENT |
| 3.5–4.0mmol/L | 52(32) | 1.00(Reference) | — | 1.00(Reference) | — |
| 4.0–4.5mmol/L | 91(40) | 1.3(0.86~1.97) | P = 0.22 | 1.28(0.84~1.96) | P = 0.26 |
| 4.5–5.0Mmol/L | 44(56) | 2.5(1.44~4.33) | P<0.01 | 2.27(1.3~3.97) | P<0.01 |
| Covariates | | | | | |
| Male | | | | 1.64(0.89~3.02) | P = 0.18 |
| Age | | | | 1.02(1.01~1.04) | P = 0.01 |
| ICU hospitalization days | | | | 0.99(0.97~1.01) | P<0.01 |
| | | Model5 | | Model6 | |
| OR (95%CI) | P-VALUE FOR TRENT | OR (95%CI) | P-VALUE FOR TRENT | | |
| Q1 | | 1.00(Reference) | — | 1.00(Reference) | — |
| Q2 | | 3.37(1.61~7.05) | P<0.01 | 2.06(1.01~4.2) | P<0.01 |
| Q3 | | 6.96(3.03~16.01) | P<0.01 | 3.24(1.43~7.36) | P<0.01 |
| Q4 | | 6.54(2.2~19.43) | P<0.01 | 2.81(1~7.89) | P<0.01 |
| Covariates | | OR (95%CI) | P-VALUE FOR TRENT | OR (95%CI) | P-VALUE FOR TRENT |
| Potassium Minimum Value | | 0.94(0.29~3.02) | P = 0.91 | 0.73(0.32~1.65) | P = 0.45 |
| Potassium Maximum Value | | 0.62(0.31~1.22) | P = 0.17 | 0.53(0.29~0.97) | P = 0.04 |
| Hyperkalemia | | 3.35(1.28~8.8) | P<0.01 | 4.98(2.34~10.61) | P<0.01 |
| Hypokalemia | | 0.86(0.4~1.86) | P = 0.71 | 0.68(0.34~1.36) | P = 0.27 |
| Potassium Average Value | | 2.19(0.66~7.29) | P = 0.2 | 2.99(1.05~8.48) | P = 0.04 |
| Male | | 1.15(0.69~1.92) | P = 0.59 | 1.11(0.67~1.82) | P = 0.69 |
| Age | | 1.02(1.01~1.04) | P<0.01 | 1.03(1.01~1.04) | P<0.01 |
| ICU hospitalization days | | 0.98(0.96~1) | P<0.01 | 0.98(0.96~1) | P<0.01 |
| SOFA | | 1.05(0.96~1.14) | P = 0.27 | 1.07(0.98~1.17) | P = 0.13 |
| Glucose SD | | 1.18(0.37~3.72) | P = 0.78 | 1.48(0.48~4.6) | P = 0.5 |
| Glucose CV | | 1.4(0~2008.21) | P = 0.93 | 1.36(0~1480.92) | P = 0.93 |
| Glucose Average | | 1.15(0.81~1.64) | P = 0.42 | 1.2(0.85~1.7) | P = 0.31 |
| Glucose maximum | | 0.84(0.61~1.14) | P = 0.25 | 0.76(0.56~1.04) | P = 0.08 |
| Glucose minimum | | 1.11(0.79~1.58) | P = 0.54 | 1.14(0.8~1.64) | P = 0.46 |
| Urine output | | 1(1~1) | P<0.01 | 1.01(1~1.07) | P<0.01 |
| Hemodialysis | | 2.27(1.22~4.22) | P<0.01 | 2.34(1.25~4.37) | P<0.01 |
| eGFR | | 1(1~1.01) | P = 0.02 | 1(1~1.01) | P = 0.03 |
| Insulin | | 1(0.99~1.01) | P = 0.56 | 1(0.99~1.01) | P = 0.58 |
| Potassium chloride | | 0.99(0.97~1) | P = 0.03 | 0.99(0.98~1) | P = 0.04 |
| Furosemide | | 1.02(1.01~1.04) | P<0.01 | 1.02(1.01~1.04) | P<0.01 |

(*Continued*)

**Table 2.** (Continued)

| PH | 4.47(0.28~71.36) | P = 0.29 | 2.81(0.17~46.94) | P = 0.47 |
|---|---|---|---|---|

Model 1: univariable model.

Model 2: multivariable model adjusted for age, male, ICU hospitalization days

Model 3: univariable model.

Model 4: multivariable model adjusted for age, male, ICU hospitalization days

Model 5: Model after adjusting for confounding factors(Potassium Minimum Value, Potassium Maximum Value, Hyperkalemia, Hypokalemia, Potassium Average Value, male, Age, ICU hospitalization days, SOFA, Glucose SD, Glucose CV, Glucose Average, Glucose maximum, Glucose minimum, Urine output, Hemodialysis, eGFR, Insulin, potassium chloride, Furosemide, PH).

Model 6: Model after multiple imputation and adjusting for confounding factors (Potassium Minimum Value, Potassium Maximum Value, Hyperkalemia, Hypokalemia, Potassium Average Value, male, Age, ICU hospitalization days, SOFA, Glucose SD, Glucose CV, Glucose Average, Glucose maximum, Glucose minimum, Urine output, Hemodialysis, eGFR, Insulin, potassium chloride, Furosemide, PH).

significant difference in mortality was found between the 3.5–4.0 mmol/L and 4–4.5 mmol/L potassium concentration groups (Table 2).

Subsequently, the samples were divided into three groups (excluding potassium>6, potassium<3, potassium>6, and potassium<3) and included in a binary logistic regression model, with CV as the independent variable and 28 day mortality rate as the dependent variable. The results were still significant (P<0.01) (Table 2 in S6 Appendix).

In Table 2, we can see that after adjusting for confounding factors and MI, significant differences were observed in gender (male), age, ICU days, mean potassium levels, hyperkalemia, urine output, Hemodialysis, eGFR, potassium chloride usage, and furosemide usage in the model (P<0.05).

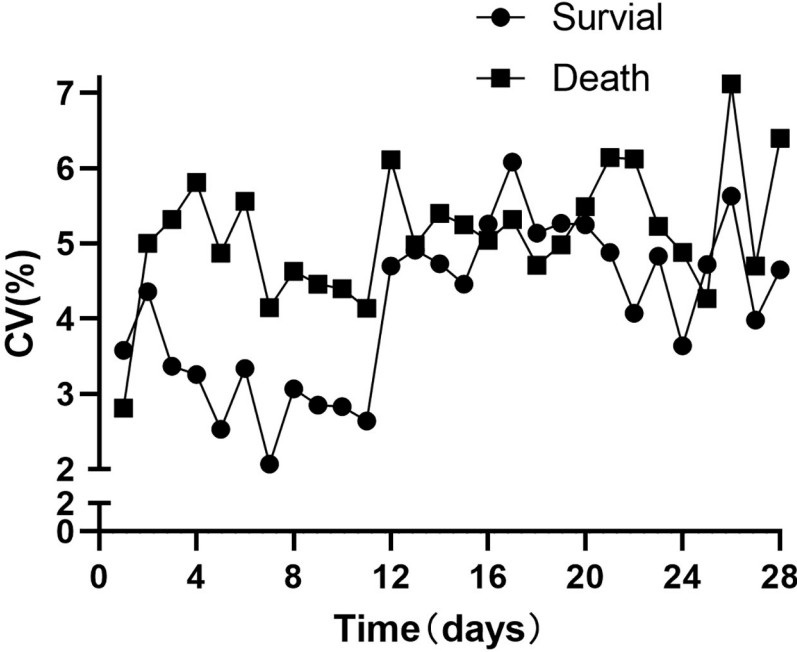

**Fig 3.**

## Discussion

Our study has identified a significant correlation between potassium ion variability and 28-day mortality rates in ICU patients. Currently, research on potassium variability is limited, and our findings align with recent studies. Lombardi et al, in their hospital-based investigation, confirmed that even when potassium ion concentration falls within normal limits, patients with heightened potassium variability confront an escalated risk of mortality [15]. Similarly, Engelhardt et al. demonstrated the beneficial effects of reducing potassium variability during ICU hospitalization [16]. Our study, differing from the cited research, visually represents potassium variability, providing insights into clinical potassium management. In a separate study, Hessels et al. employed computer-assisted control to manage hyperkalemia, hypokalemia, and potassium variability. Significantly, this study found an independent association between potassium variability and mortality [17]. Currently, specific guidelines for potassium management in ICU patients are lacking. Regarding the optimal potassium concentration for ICU patients, studies report inconsistent ranges, fluctuating between 3.5–5.5 mmol/L [16, 18–21]. Additionally, there is variability in the recommended quantity and speed of potassium supplementation for hypokalemia, and intravenous potassium administration often results in hyperkalemia [9, 22–26]. Therefore, personalized approaches for potassium supplementation and reduction may be necessary [27]. While our study highlights the correlation between potassium ion variability and 28-day mortality in ICU patients, it does not establish a causal relationship. Some research suggests diurnal variations in potassium concentration, with higher day-night differences observed in deceased patients [4, 28]. Moreover, ICU patients often consume potassium in excess of theoretical values [29], indicating an unstable internal environment and the severity of their condition.

Additionally, factors such as insulin dosage, potassium chloride dosage, average blood glucose, and furosemide dosage may influence potassium concentration [30]. Higher insulin levels are associated with increased cellular absorption of potassium. Intravenous injection of insulin can increase intracellular sodium sensitivity, activate Na+- K+ATPase, and inhibition of potassium efflux [31]. Furosemide is often used to reduce fluid overload in patients, but it also often leads to hypokalemia. The mechanism by which furosemide reduces blood potassium is by inhibiting sodium reabsorption in the renal tubules, increasing the transmission of sodium in aldosterone sensitive distal tubules, and thereby stimulating potassium secretion [32]. Regarding the relationship between blood sugar and blood potassium, studies have shown that the mechanism by which blood sugar reduces blood potassium may involve hypokalemia caused by diuretics. In addition, hypokalemia may lead to glucose intolerance, i.e. elevated blood sugar levels [33].

Our model suggests that the longer the ICU days and the older the age, the higher the patient mortality rate. Due to the presence of potassium in urine output and its correlation with eGFR, both urine output and eGFR can affect the magnitude of potassium variability. During hemodialysis, potassium in the dialysate is exchanged with potassium in the blood through a semi permeable membrane. The potassium concentration in dialysate is usually lower than that in blood, so potassium will transfer from blood to dialysate. During this process, potassium is removed, which may lead to changes in potassium variability.

ICU patients frequently present with multiple diseases, and the emergence of sepsis, delirium, and ICU-acquired muscle weakness in the ICU may compromise patient survival or quality of life [34, 35]. Two prospective studies identified multi-organ dysfunction as the primary cause of death during ICU hospitalization, while malignancies and chronic cardiovascular diseases were the main causes of death post-discharge [36, 37]. Consequently, a one-size-fits-all approach to determining the cause of death in patients is not applicable.

To definitively establish a causal relationship between potassium variability and mortality, additional well-designed randomized controlled trials (RCTs) are essential. These studies must consider the numerous factors within the ICU environment that influence potassium levels and patient outcomes. Currently, it is established that both excessively high and critically low potassium concentrations can elevate the risk of mortality [38]. Clinical practice typically concentrates solely on potassium concentration, neglecting the amplitude and variability of these changes. Therefore, when potassium levels are within the safe range, assessing potassium variability could predict the short-term (28-day) risk of mortality. Should a patient be predicted at high risk, appropriate clinical interventions, such as modifying the regimen of potassium supplementation or reduction, may be undertaken to mitigate mortality risk. However, whether such interventions effectively reduce mortality rates remains debatable. This is because our study only observed an association between high potassium variability and 28-day mortality.

In addition, we noticed that gender (male) is significantly present in the statistical model, relevant literature points out that male or female exposure may affect the incidence rate and prognosis of various critical diseases throughout the life cycle. Biological differences, sex hormone and cytokine responses may lead to observed gender differences, but the mechanisms underlying these gender differences cannot currently be explained [39].

Consequently, further research focusing on control variables is imperative. In clinical treatment, precise timing and appropriate dosing of potassium supplementation are necessary. This aspect is currently under-addressed. Addressing the interplay between potassium variability and mortality could foster more targeted and efficacious clinical interventions, ultimately enhancing outcomes for ICU patients.

## Advantages and limitations of the study

### Advantages

In clinical practice, the focus is often solely on the concentration of potassium, neglecting the amplitude and range of its fluctuations. Our study demonstrates that high variability in potassium levels is positively correlated with increased 28-day mortality. Furthermore, by collecting potassium concentration values every 12 hours, we within ICU 28 days, were able to more effectively monitor the amplitude and variability of potassium CV over time. Additionally, we plotted the trends in potassium concentration and its coefficient of variation (CV) over 28 days, facilitating a clearer observation of how potassium CV changes under various conditions.

### Limitations

Our study, being a retrospective, single-center survey, inherently limits the generalizability of our conclusions. Additionally, the methods and dosages of medications administered during hospitalization were not systematically recorded. Given the wide variety of drugs utilized, electrolyte imbalances frequently occur in ICU patients [40]. Although data were collected on medications administered on the first day of hospitalization that could influence potassium concentrations, daily dosages and dosing rates were not recorded throughout the hospital stay. While potassium concentrations were measured at 12-hour intervals, the study lacked information on the occurrence of hyperkalemia or hypokalemia during these intervals. Current consensus guidelines recommend assessing potassium concentrations every 2–4 hours [41].

## Conclusion

High potassium variability in ICU patients is a significant risk factor for increased 28-day mortality.

## Supporting information

**S1 Appendix.**
(PDF)

**S2 Appendix. Additional description of missing potassium measures.**
(PDF)

**S3 Appendix. Comparison of potassium variability grouping and potassium concentration grouping.**
(PDF)

**S4 Appendix. Changes in potassium concentration or CV within 28 days.**
(PDF)

**S5 Appendix. General information and comparison of death and survival groups.**
(PDF)

**S6 Appendix. Binary logistic regression model and normality test and single factor analysis.**
(PDF)

## Author Contributions

**Conceptualization:** ShengDe Liang.

**Data curation:** YuChou Zhang.

**Validation:** HanChun Wen.

**Visualization:** HanChun Wen.

**Writing – original draft:** YuChou Zhang.

**Writing – review & editing:** YuChou Zhang.

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
