## [Decision Letter · Decision Letter 0]

2 Jul 2024

PONE-D-24-19608The Impact of Serum Potassium Ion Variability on 28-Day Mortality in ICU PatientsPLOS ONE

Dear Dr. Zhang,

Thank you for submitting your manuscript to PLOS ONE. After careful consideration, we feel that it has merit but does not fully meet PLOS ONE’s publication criteria as it currently stands. Therefore, we invite you to submit a revised version of the manuscript that addresses the points raised during the review process.

Two experts raised several concerns to be clarified.

We look forward to receiving your revised manuscript.

Kind regards,

Tatsuo Shimosawa, M.D., Ph.D.

Academic Editor

PLOS ONE

Journal Requirements:

Reviewers' comments:

Reviewer's Responses to Questions

**Comments to the Author**

1. Is the manuscript technically sound, and do the data support the conclusions?

Reviewer #1: Yes

Reviewer #2: Yes

2. Has the statistical analysis been performed appropriately and rigorously? 

Reviewer #1: I Don't Know

Reviewer #2: Yes

3. Have the authors made all data underlying the findings in their manuscript fully available?

Reviewer #1: Yes

Reviewer #2: Yes

4. Is the manuscript presented in an intelligible fashion and written in standard English?

Reviewer #1: Yes

Reviewer #2: Yes

5. Review Comments to the Author

Reviewer #1: Authors have investigated the correlation between variability of serum K and mortality and have found that the higher variability was associated with higher morality. I found the results very interesting and novel, which merits publication. However, some comments needs to be answered before acceptance.

1) Since it is well known that mortality is associated with extreme values of sK, in order to exclude the effect of severe hypo- of heperkalemia in terms of mortality, I would recommend to check the correlation between CV and moratality only in those without experiencing extreme values (for example, sK<3, >6) in addition to whole analysis. This analysis should serve as a sensitivity analysis.

2) Significant change in sK can occur especially in those with hemodialysis or CRRT. I would recommend to add renal replacement therapy (yes or no) as a variable in the analysis.

3) Also, sK change can be exaggerated in those with lreduced kidney function. It looks like "renal insufficiency" is rare in the study population, how you define "renal insufficiency"? I would like to see the effect of degree of eGFR to the CV of sK and also to the mortality.

Reviewer #2: In patients requiring ICU management, the best relationship between death and K value is around 4.0, as previously reported. The paper also attempts to prove that a lower coefficient of variation of K is associated with an increased risk of 28-day mortality, which is an interesting point of view. However, in order to prove this, the following issues need to be resolved.

Major comment

①Overall, the correlation between K values and mortality has been reported in large clinical studies.

The total number of patients is 237, which is too small to statistically correlate. It is necessary to increase the number of patients to examine the correlation.

➁The period of observation is slightly different and more detailed data is needed to correlate the one-week variation in CV of K-values with the risk of death during 28 days. (If possible, K values and CV values should also be listed for up to 28 days.)

③The K-adjustment method for dialysis patients is different from that for other patients, so separate analyses are recommended.

Minor comment

①In Table 2 Covariates, P value is described from Male to PH, but OR (95%) is described. Confirmation is needed.

➁It is necessary to list the mean K values before admission to ICU and at 28 days.

③The report concludes that there is a significant difference by sex (male), but it is necessary to explain the reason for this in the discussion.

6. PLOS authors have the option to publish the peer review history of their article (what does this mean?). If published, this will include your full peer review and any attached files.

Reviewer #1: No

Reviewer #2: No

---

## [Author Response · Author response to Decision Letter 0]

1 Aug 2024

We thank the reviewers for their valuable comments, which certainly help us to 

improve the quality of our manuscript. All comments are addressed on a point-by-point 

basis below, where letters C & R denote Comment and Response, respectively. 

Reviewer#1: 

C:1)Since it is well known that the portality is associated with extreme values of sK,in order to exclude the effect of evere hypo-ofhepercalemia and termsofortality, I would recommend checking the correlation between CV and moratality only in those without experiencing extreme values(for example,sK<3,>6)inaddition to wholel analysis.This analysis should serve as a sensitivy analysis.

R:1)Thank you for your feedback, I agree with your point of view. Therefore, I established three statistical models. Patients with K>6, K<3, K>6, and K<3 were excluded and reanalyzed. Table 2 in Appendix 6 shows that all three statistical models are significant.

C:2)Significant change in sK can occur especially in those with hemodialysis or CRRT. I would recommend to add renal replacement therapy (yes or no) as a variable in the analysis.

R:2)Thank you for your suggestion. I agree with your viewpoint. We added the variable of hemodialysis to the statistical model and found P<0.01 (model 4 in Appendix 6). Moreover, after excluding patients undergoing hemodialysis and reanalyzing, it was found to still be significant (Table 4 in Appendix 6).

C:3)Also, sK change can be exaggerated in those with lreduced kidney function. It looks like "renal insufficiency" is rare in the study population, how you define "renal insufficiency"? I would like to see the effect of degree of eGFR to the CV of sK and also to the mortality.

R:3)Thank you for your valuable feedback. We have made improvements based on your feedback. ICD-10-diagnoses were determined by physicians depending on the guidelines valid at the time. ICD-10-GM derived covariates are as followed: Renal Insufficiency N18.001-N18.004. eGFR was calculated based on age, gender, and creatinine levels. After incorporating the covariate eGFR into the statistical model, it showed significant differences (P=0.03) (Model 5). In addition, Appendix 5 shows a significant difference in eGFR between the death group and the inventory group.

Reviewer#2

C:1）Overall, the correlation between K values and mortality has been reported in large clinical studies.The total number of patients is 237, which is too small to statistically correlate. It is necessary to increase the number of patients to examine the correlation.

R:1）Thank you for your suggestion. We agree with your opinion, so we have increased the sample size and reanalyzed the article.

C:2）The period of observation is slightly different and more detailed data is needed to correlate the one-week variation in CV of K-values with the risk of death during 28 days. (If possible, K values and CV values should also be listed for up to 28 days.) 

R:2）Thank you for your suggestion. We agree with your viewpoint. We increased the observation period of K to 28 days, and then collected relevant data and reanalyzed it.

C:3）The K-adjustment method for dialysis patients is different from that for other patients, so separate analyses are recommended.

R:3）We agree with your viewpoint. Therefore, we excluded patients who had undergone blood purification from the statistical model and conducted reanalysis, and the results still showed significance (Table 4 in Appendix 6)

C:4）In Table 2 Covariates, P value is described from Male to PH, but OR (95%) is described.

Confirmation is needed.

R:4）Thank you for your feedback. Table 2 describes OR (95% CI).

C:5）It is necessary to list the mean K values before admission to ICU and at 28 days.

R:5）Thank you for your suggestion. We collected potassium values within 28 days of ICU and reanalyzed them.

C:6）The report concludes that there is a significant difference by sex (male), but it is necessary to

explain the reason for this in the discussion.

R:6）We agree with your opinion and have made revisions in the article. For the significance of men in the statistical model, relevant literature points out that male or female exposure may affect the incidence rate and prognosis of various critical diseases throughout the life cycle. Biological differences, sex hormone and cytokine responses may lead to observed gender differences, but the mechanisms underlying these gender differences cannot currently be explained.

---

## [Decision Letter · Decision Letter 1]

16 Aug 2024

The Impact of Serum Potassium Ion Variability on 28-Day Mortality in ICU Patients

PONE-D-24-19608R1

Dear Dr. Zhang,

We’re pleased to inform you that your manuscript has been judged scientifically suitable for publication and will be formally accepted for publication once it meets all outstanding technical requirements.

Kind regards,

Tatsuo Shimosawa, M.D., Ph.D.

Academic Editor

PLOS ONE

Additional Editor Comments (optional):

Reviewers' comments:

Reviewer's Responses to Questions

**Comments to the Author**

1. If the authors have adequately addressed your comments raised in a previous round of review and you feel that this manuscript is now acceptable for publication, you may indicate that here to bypass the “Comments to the Author” section, enter your conflict of interest statement in the “Confidential to Editor” section, and submit your "Accept" recommendation.

Reviewer #1: All comments have been addressed

Reviewer #2: All comments have been addressed

2. Is the manuscript technically sound, and do the data support the conclusions?

Reviewer #1: Yes

Reviewer #2: Yes

3. Has the statistical analysis been performed appropriately and rigorously? 

Reviewer #1: I Don't Know

Reviewer #2: Yes

4. Have the authors made all data underlying the findings in their manuscript fully available?

Reviewer #1: Yes

Reviewer #2: Yes

5. Is the manuscript presented in an intelligible fashion and written in standard English?

Reviewer #1: Yes

Reviewer #2: Yes

6. Review Comments to the Author

Reviewer #1: All comments are fairly well answered and appropriate revisions have been made. I have no further comments.

Reviewer #2: (No Response)

7. PLOS authors have the option to publish the peer review history of their article (what does this mean?). If published, this will include your full peer review and any attached files.

Reviewer #1: No

Reviewer #2: No

---

## [Editor Report · Acceptance letter]

30 Aug 2024

PONE-D-24-19608R1 

PLOS ONE

Dear Dr. Zhang, 

I'm pleased to inform you that your manuscript has been deemed suitable for publication in PLOS ONE. Congratulations! Your manuscript is now being handed over to our production team.

Kind regards, 

on behalf of

Prof. Tatsuo Shimosawa 

Academic Editor

PLOS ONE